# Towards better understanding of gradient-based attribution methods for Deep Neural Networks

**Marco Ancona**
Department of Computer Science
ETH Zurich, Switzerland
`marco.ancona@inf.ethz.ch`

**Enea Ceolini**
Institute of Neuroinformatics
University Zürich and ETH Zürich
`enea.ceolini@ini.uzh.ch`

**Cengiz Öztireli**
Department of Computer Science
ETH Zurich, Switzerland
`cengizo@inf.ethz.ch`

**Markus Gross**
Department of Computer Science
ETH Zurich, Switzerland
`grossm@inf.ethz.ch`

## Abstract

Understanding the flow of information in Deep Neural Networks (DNNs) is a challenging problem that has gain increasing attention over the last few years. While several methods have been proposed to explain network predictions, there have been only a few attempts to compare them from a theoretical perspective. What is more, no exhaustive empirical comparison has been performed in the past. In this work, we analyze four gradient-based attribution methods and formally prove conditions of equivalence and approximation between them. By reformulating two of these methods, we construct a unified framework which enables a direct comparison, as well as an easier implementation. Finally, we propose a novel evaluation metric, called *Sensitivity-n* and test the gradient-based attribution methods alongside with a simple perturbation-based attribution method on several datasets in the domains of image and text classification, using various network architectures.

## 1 Introduction and Motivation

While DNNs have had a large impact on a variety of different tasks (LeCun et al., 2015; Krizhevsky et al., 2012; Mnih et al., 2015; Silver et al., 2016; Wu et al., 2016), explaining their predictions is still challenging. The lack of tools to inspect the behavior of these black-box models makes DNNs less trustable for those domains where interpretability and reliability are crucial, like autonomous driving, medical applications and finance.

In this work, we study the problem of assigning an *attribution* value, sometimes also called "relevance" or "contribution", to each input feature of a network. More formally, consider a DNN that takes an input $x = [x_1, ..., x_N] \in \mathbb{R}^N$ and produces an output $S(x) = [S_1(x), ..., S_C(x)]$, where $C$ is the total number of output neurons. Given a specific target neuron $c$, the goal of an attribution method is to determine the contribution $R^c = [R_1^c, ..., R_N^c] \in \mathbb{R}^N$ of each input feature $x_i$ to the output $S_c$. For a classification task, the target neuron of interest is usually the output neuron associated with the correct class for a given sample. When the attributions of all input features are arranged together to have the same shape of the input sample we talk about *attribution maps* (Figures 1-2), which are usually displayed as heatmaps where red color indicates features that contribute positively to the activation of the target output, and blue color indicates features that have a suppressing effect on it.

The problem of finding attributions for deep networks has been tackled in several previous works (Simonyan et al., 2014; Zeiler & Fergus, 2014; Springenberg et al., 2014; Bach et al., 2015; Shrikumar et al., 2017; Sundararajan et al., 2017; Montavon et al., 2017; Zintgraf et al., 2017). Unfortunately, due to slightly different problem formulations, lack of compatibility with the variety of existing DNN architectures and no common benchmark, a comprehensive comparison is not available. Various

new attribution methods have been published in the last few years but we believe a better theoretical understanding of their properties is fundamental. The contribution of this work is twofold:

1. We prove that $\epsilon$-LRP (Bach et al., 2015) and DeepLIFT (Rescale) (Shrikumar et al., 2017) can be reformulated as computing backpropagation for a modified gradient function (Section 3). This allows the construction of a unified framework that comprises several gradient-based attribution methods, which reveals how these methods are strongly related, if not equivalent under certain conditions. We also show how this formulation enables a more convenient implementation with modern graph computational libraries.

2. We introduce the definition of *Sensitivity-n*, which generalizes the properties of *Completeness* (Sundararajan et al., 2017) and *Summation to Delta* (Shrikumar et al., 2017) and we compare several methods against this metric on widely adopted datasets and architectures. We show how empirical results support our theoretical findings and propose directions for the usage of the attribution methods analyzed (Section 4).

## 2 OVERVIEW OVER EXISTING ATTRIBUTION METHODS

### 2.1 PERTURBATION-BASED METHODS

Perturbation-based methods directly compute the attribution of an input feature (or set of features) by removing, masking or altering them, and running a forward pass on the new input, measuring the difference with the original output. This technique has been applied to Convolutional Neural Networks (CNNs) in the domain of image classification (Zeiler & Fergus, 2014), visualizing the probability of the correct class as a function of the position of a grey patch occluding part of the image. While perturbation-based methods allow a direct estimation of the marginal effect of a feature, they tend to be very slow as the number of features to test grows (ie. up to hours for a single image (Zintgraf et al., 2017)). What is more, given the nonlinear nature of DNNs, the result is strongly influenced by the number of features that are removed altogether at each iteration (Figure 1).

In the remainder of the paper, we will consider the occluding method by Zeiler & Fergus (2014) as a comparison benchmark for perturbation-based methods. We will use this method, referred to as *Occlusion-1*, replacing one feature $x_i$ at the time with a zero baseline and measuring the effect of this perturbation on the target output, ie. $S_c(x) - S_c(x_{[x_i=0]})$ where we use $x_{[x_i=v]}$ to indicate a sample $x \in R^N$ whose $i$-th component has been replaced with $v$. The choice of zero as a baseline is consistent with the related literature and further discussed in Appendix B.

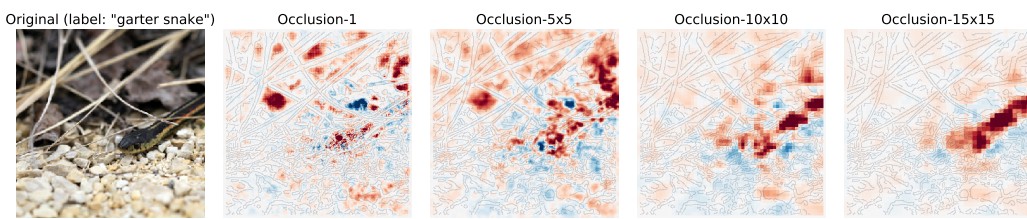

Figure 1: Attributions generated by occluding portions of the input image with squared grey patches of different sizes. Notice how the size of the patches influence the result, with focus on the main subject only when using bigger patches.

### 2.2 BACKPROPAGATION-BASED METHODS

Backpropagation-based methods compute the attributions for all input features in a single forward and backward pass through the network [1]. While these methods are generally faster then perturbation-based methods, their outcome can hardly be directly related to a variation of the output.

---

[1] Sometimes several of these steps are necessary, but the number does not depend on the number of input feature and generally much smaller than for perturbation-based methods

**Gradient * Input** (Shrikumar et al., 2016) was at first proposed as a technique to improve the sharpness of the attribution maps. The attribution is computed taking the (signed) partial derivatives of the output with respect to the input and multiplying them with the input itself. Refer to Table 1 for the mathematical definition.

**Integrated Gradients** (Sundararajan et al., 2017), similarly to Gradient * Input, computes the partial derivatives of the output with respect to each input feature. However, while Gradient * Input computes a single derivative, evaluated at the provided input $x$, Integrated Gradients computes the *average* gradient while the input varies along a linear path from a baseline $\bar{x}$ to $x$. The baseline is defined by the user and often chosen to be zero. We report the mathematical definition in Table 1.

Integrated Gradients satisfies a notable property: the attributions sum up to the target output minus the target output evaluated at the baseline. Mathematically, $\sum_{i=1}^{N} R_i^c(x) = S_c(x) - S_c(\bar{x})$. In related literature, this property has been variously called *Completeness* (Sundararajan et al., 2017), *Summation to Delta* (Shrikumar et al., 2017) or *Efficiency* in the context of cooperative game theory (Roth, 1988), and often recognized as desirable for attribution methods.

**Layer-wise Relevance Propagation (LRP)** (Bach et al., 2015) is computed with a backward pass on the network. Let us consider a quantity $r_i^{(l)}$, called "relevance" of unit $i$ of layer $l$. The algorithm starts at the output layer $L$ and assigns the relevance of the target neuron $c$ equal to the output of the neuron itself and the relevance of all other neurons to zero (Eq. 1).

The algorithm proceeds layer by layer, redistributing the prediction score $S_i$ until the input layer is reached. One recursive rule for the redistribution of a layer's relevance to the following layer is the $\epsilon$-rule described in Eq. 2, where we defined $z_{ji} = w_{ji}^{(l+1,l)} x_i^{(l)}$ to be the weighted activation of a neuron $i$ onto neuron $j$ in the next layer and $b_j$ the additive bias of unit $j$. A small quantity $\epsilon$ is added to the denominator of Equation 2 to avoid numerical instabilities. Once reached the input layer, the final attributions are defined as $R_i^c(x) = r_i^{(1)}$.

$$r_i^{(L)} = \begin{cases} S_i(x) & \text{if unit } i \text{ is the target unit of interest} \\ 0 & \text{otherwise} \end{cases} \tag{1}$$

$$r_i^{(l)} = \sum_j \frac{z_{ji}}{\sum_{i'}(z_{ji'} + b_j) + \epsilon \cdot sign(\sum_{i'}(z_{ji'} + b_j))} r_j^{(l+1)} \tag{2}$$

LRP together with the propagation rule described in Eq. 2 is called $\epsilon$-LRP, analyzed in the remainder of this paper. There exist alternative stabilizing methods described in Bach et al. (2015) and Montavon et al. (2017) which we do not consider here.

**DeepLIFT** (Shrikumar et al., 2017) proceeds in a backward fashion, similarly to LRP. Each unit $i$ is assigned an attribution that represents the relative effect of the unit activated at the original network input $x$ compared to the activation at some reference input $\bar{x}$ (Eq. 3). Reference values $\bar{z}_{ji}$ for all hidden units are determined running a forward pass through the network, using the baseline $\bar{x}$ as input, and recording the activation of each unit. As in LRP, the baseline is often chosen to be zero. The relevance propagation is described in Eq. 4. The attributions at the input layer are defined as $R_i^c(x) = r_i^{(1)}$ as for LRP.

$$r_i^{(L)} = \begin{cases} S_i(x) - S_i(\bar{x}) & \text{if unit } i \text{ is the target unit of interest} \\ 0 & \text{otherwise} \end{cases} \tag{3}$$

$$r_i^{(l)} = \sum_j \frac{z_{ji} - \bar{z}_{ji}}{\sum_{i'} z_{ji} - \sum_{i'} \bar{z}_{ji}} r_j^{(l+1)} \tag{4}$$

In Equation 4, $\bar{z}_{ji} = w_{ji}^{(l+1,l)} \bar{x}_i^{(l)}$ is the weighted activation of a neuron $i$ onto neuron $j$ when the baseline $\bar{x}$ is fed into the network. As for Integrated Gradients, DeepLIFT was designed to satisfy *Completeness*. The rule described in Eq. 4 ("Rescale rule") is used in the original formulation of the method and it is the one we will analyze in the remainder of the paper. The "Reveal-Cancel" rule (Shrikumar et al., 2017) is not considered here.

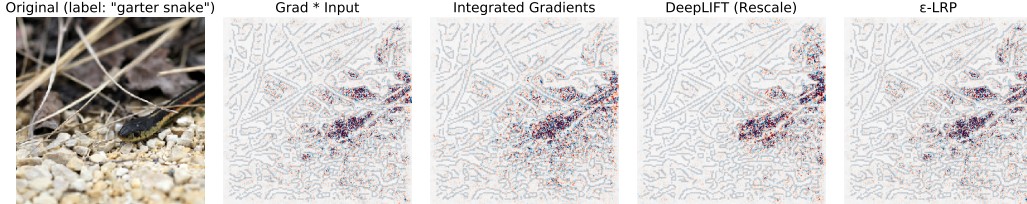

Figure 2: Attribution generated by applying several attribution methods to an Inception V3 network for natural image classification (Szegedy et al., 2016). Notice how all gradient-based methods produce attributions affected by higher local variance compared to perturbation-based methods (Figure 1).

Other back-propagation methods exist. **Saliency maps** (Simonyan et al., 2014) constructs attributions by taking the absolute value of the partial derivative of the target output $S_c$ with respect to the input features $x_i$. Intuitively, the *absolute value* of the gradient indicates those input features (pixels, for image classification) that can be perturbed the least in order for the target output to change the most. However, the absolute value prevents the detection of positive and negative evidence that might be present in the input, reason for which this method will not be used for comparison in the remainder of the paper. Similarly, **Deep Taylor Decomposition** (Montavon et al., 2017), although showed to produce sparser explanations, assumes no negative evidence in the input and produces only positive attribution maps. We show in Section 4 that this assumption does not hold for our tasks. Other methods that are designed only for specific architectures (ie. **Grad-CAM** (Selvaraju et al., 2016) for CNNs) or activation functions (ie. **Deconvolutional Network** (Zeiler & Fergus, 2014) and **Guided Backpropagation** (Springenberg et al., 2014) for ReLU) are also out of the scope of this analysis, since our goal is to perform a comparison across multiple architectures and tasks.

## 3 A UNIFIED FRAMEWORK

Gradient * Input and Integrated Gradients are, by definition, computed as a function of the partial derivatives of the target output with respect to each input feature. In this section, we will show that $\epsilon$-LRP and DeepLIFT can also be computed by applying the chain rule for gradients, if the instant gradient at each nonlinearity is replaced with a function that depends on the method.

In a DNN where each layer performs a linear transformation $z_j = \sum_i w_{ji} x_i + b_j$ followed by a nonlinear mapping $x_j = f(z_j)$, a path connecting any two units consists of a sequence of such operations. The chain rule along a single path is therefore the product of the partial derivatives of all linear and nonlinear transformations along the path. For two units $i$ and $j$ in *subsequent* layers we have $\partial x_j / \partial x_i = w_{ji} \cdot f'(z_j)$, whereas for any two generic units $i$ and $c$ connected by a set of paths $P_{ic}$ the partial derivative is sum of the product of all weights $w_p$ and all derivatives of the nonlinearities $f'(z)_p$ along each path $p \in P_{ic}$. We introduce a notation to indicate a modified chain-rule, where the derivative of the nonlinearities $f'()$ is replaced by a generic function $g()$:

$$\frac{\partial^g x_c}{\partial x_i} = \sum_{p \in P_{ic}} \left( \prod w_p \prod g(z)_p \right) \tag{5}$$

When $g() = f'()$ this is the definition of partial derivative of the output of unit $c$ with respect to unit $i$, computed as the sum of contributions over all paths connecting the two units. Given that a zero weight can be used for non-existing or blocked paths, this is valid for any architecture that involves fully-connected, convolutional or recurrent layers without multiplicative units, as well as for pooling operations.

**Proposition 1.** *$\epsilon$-LRP is equivalent the feature-wise product of the input and the modified partial derivative $\partial^g S_c(x)/\partial x_i$, with $g = g^{LRP} = f_i(z_i)/z_i$, i.e. the ratio between the output and the input at each nonlinearity.*

**Proposition 2.** *DeepLIFT (Rescale) is equivalent to the feature-wise product of the $x - \bar{x}$ and the modified partial derivative $\partial^g S_c(x)/\partial x_i$, with $g = g^{DL} = (f_i(z_i) - f_i(\bar{z}_i))/(z_i - \bar{z}_i)$, i.e. the*

*ratio between the difference in output and the difference in input at each nonlinearity, for a network provided with some input $x$ and some baseline input $\bar{x}$ defined by the user.*

The proof for Proposition 1 and 2 are provided in Appendix A.1 and Appendix A.2 respectively. Given these results, we can write all methods with a consistent notation. Table 1 summaries the four methods considered and shows examples of attribution maps generated by these methods on MNIST.

| Method | Attribution $R_i^c(x)$ | Example of attributions on MNIST |
|---|---|---|
| | | ReLU  Tanh  Sigmoid  Softplus |
| Gradient * Input | $x_i \cdot \dfrac{\partial S_c(x)}{\partial x_i}$ |  |
| Integrated Gradient | $(x_i - \bar{x}_i) \cdot \displaystyle\int_{\alpha=0}^{1} \left. \dfrac{\partial S_c(\tilde{x})}{\partial(\tilde{x}_i)}\right|_{\tilde{x}=\bar{x}+\alpha(x-\bar{x})} d\alpha$ |  |
| $\epsilon$-LRP | $x_i \cdot \dfrac{\partial^g S_c(x)}{\partial x_i}, \quad g = \dfrac{f(z)}{z}$ |  |
| DeepLIFT | $(x_i - \bar{x}_i) \cdot \dfrac{\partial^g S_c(x)}{\partial x_i}, \quad g = \dfrac{f(z) - f(\bar{z})}{z - \bar{z}}$ |  |
| Occlusion-1 | $S_c(x) - S_c(x_{[x_i=0]})$ |  |

Table 1: Mathematical formulation of five gradient-based attribution methods and of Occlusion-1. The formulation for the two underlined methods is derived from Propositions 1-2. On the right, examples of attributions on the MNIST dataset (LeCun et al., 1998) with four CNNs using different activation functions. Details on the architectures can be found in Appendix C.

As pointed out by Sundararajan et al. (2017) a desirable property for attribution methods is their immediate applicability to existing models. Our formulation makes this possible for $\epsilon$-LRP and DeepLIFT. Since all modern frameworks for graph computation, like the popular TensorFlow (Abadi et al., 2015), implement backpropagation for efficient computation of the chain rule, it is possible to implement all methods above by the gradient of the graph nonlinearities, with no need to implement custom layers or operations. Listing 1 shows an example of how to achieve this on Tensorflow.

```
1  @ops.RegisterGradient("GradLRP")
2  def _GradLRP(op, grad):
3      op_out = op.outputs[0]
4      op_in = op.inputs[0]
5      return grad * op_out / (op_in + eps)
```

Listing 1: Example of gradient override for a Tensorflow operation. After registering this function as the gradient for nonlinear activation functions, a call to `tf.gradients()` and the multiplication with the input will produce the $\epsilon$-LRP attributions.

### 3.1 INVESTIGATING FURTHER CONNECTIONS

The formulation of Table 1 facilitates the comparison between these methods. Motivated by the fact that attribution maps for different gradient-based methods look surprisingly similar on several tasks, we investigate some conditions of equivalence or approximation.

**Proposition 3.** *$\epsilon$-LRP is equivalent to i) Gradient * Input if only Rectified Linear Units (ReLUs) are used as nonlinearities; ii) DeepLIFT (computed with a zero baseline) if applied to a network with no additive biases and with nonlinearities $f$ such that $f(0) = 0$ (eg. ReLU or Tanh).*

The first part of Proposition 3 comes directly as a corollary of Proposition 1 by noticing that for ReLUs the gradient at the nonlinearity $f'$ is equal to $g^{LRP}$ for all inputs. This relation has been previously proven by Shrikumar et al. (2016) and Kindermans et al. (2016). Similarly, we notice that, in a network with no additive biases and nonlinearities that cross the origin, the propagation of the baseline produces a zero reference value for *all* hidden units (ie. $\forall i : \bar{z}_i = f(\bar{z}_i) = 0$). Then $g^{LRP} = g^{DL}$, which proves the second part of the proposition.

Notice that $g^{LRP}(z) = (f(z) - 0)/(z - 0)$ which, in the case of ReLU and Tanh, is the *average gradient* of the nonlinearity in $[0, z]$. It also easy to see that $\lim_{z \to 0} g^{LRP}(z) = f'(0)$, which explain why $g$ can not assume arbitrarily large values as $z \to 0$, even without stabilizers. On the contrary, if the discussed condition on the nonlinearity is not satisfied, for example with Sigmoid or Softplus, we found empirically that $\epsilon$-LRP fails to produce meaningful attributions as shown in the empirical comparison of Section 4. We speculate this is due to the fact $g^{LRP}(z)$ can become extremely large for small values of $z$, being its upper-bound only limited by the stabilizer. This causes attribution values to concentrate on a few features as shown in Table 1. Notice also that the interpretation of $g^{LRP}$ as average gradient of the nonlinearity does not hold in this case, which explains why $\epsilon$-LRP diverges from other methods [2].

DeepLIFT and Integrated Gradients are related as well. While Integrated Gradients computes the average partial derivative of each feature as the input varies from a baseline to its final value, DeepLIFT approximates this quantity in a single step by replacing the gradient at each nonlinearity with its average gradient. Although the chain rule does not hold in general for average gradients, we show empirically in Section 4 that DeepLIFT is most often a good approximation of Integrated Gradients. This holds for various tasks, especially when employing simple models (see Figure 4). However, we found that DeepLIFT diverges from Integrated Gradients and fails to produce meaningful results when applied to Recurrent Neural Networks (RNNs) with multiplicative interactions (eg. gates in LSTM units (Hochreiter & Schmidhuber, 1997)). With multiplicative interactions, DeepLIFT does not satisfy *Completeness*, which can be illustrated with a simple example. Take two variables $x_1$ and $x_2$ and a the function $h(x_1, x_2) = ReLU(x_1 - 1) \cdot ReLU(x_2)$. It can be easily shown that, by applying the methods as described by Table 1, DeepLIFT does not satisfy *Completeness*, one of its fundamental design properties, while Integrated gradients does.

## 3.2 Local and Global attribution methods

The formulation in Table 1 highlights how all the gradient-based methods considered are computed from a quantity that depends on the weights and the architecture of the model, multiplied by the input itself. Similarly, Occlusion-1 can also be interpreted as the input multiplied by the *average* value of the partial derivatives, computed varying one feature at the time between zero and their final value:

$$R_i^c(x) = S_c(x) - S_c(x_{[x_i=0]}) = x_i \cdot \int_{\alpha=0}^{1} \frac{\partial S_c(\tilde{x})}{\partial(\tilde{x}_i)} \bigg|_{\tilde{x}=x_{[x_i=\alpha \cdot x_i]}} d\alpha$$

The reason justifying the multiplication with the input has been only partially discussed in previous literature (Smilkov et al., 2017; Sundararajan et al., 2017; Shrikumar et al., 2016). In many cases, it contributes to making attribution maps sharper although it remains unclear how much of this can be attributed to the sharpness of the original image itself. We argue the multiplication with the input has a more fundamental justification, which allows to distinguish attribution methods in two broad categories: *global attribution methods*, that describe the marginal effect of a feature on the output with respect to a baseline and; *local attribution methods*, that describe how the output of the network changes for infinitesimally small perturbations around the original input.

For a concrete example, we will consider the linear case. Imagine a linear model to predict the total capital in ten years $C$, based on two investments $x_1$ and $x_2$: $C = 1.05 \cdot x_1 + 10 \cdot x_2$. Given this

---

[2]We are not claiming any general superiority of gradient-based methods but rather observing that $\epsilon$-LRP can only be considered gradient-based for precise choices of the nonlinearities. In fact, there are backpropagation-based attribution methods, not directly interpretable as gradient methods, that exhibit other desirable properties. For a discussion about advantages and drawbacks of gradient-based methods we refer the reader to Shrikumar et al. (2017); Montavon et al. (2018); Sundararajan et al. (2017).

simple model, $R_1 = \partial C / \partial x_1 = 1.05$, $R_2 = \partial C / \partial x_2 = 10$ represents a possible local attribution. With no information about the actual value of $x_1$ and $x_2$ we can still answer the question *"Where should one invest in order to generate more capital?*. The local attributions reveal, in fact, that by investing $x_2$ we will get about ten times more return than investing in $x_1$. Notice, however, that this does not tell anything about the contribution to the total capital for a specific scenario. Assume $x_1 = 100'000\$$ and $x_2 = 1'000\$$. In this scenario $C = 115000\$$. We might ask ourselves *"How the initial investments contributed to the final capital?"*. In this case, we are looking for a global attribution. The most natural solution would be $R_1 = 1.05x_1 = 105'000\$$, $R_2 = 10x_2 = 1'000\$$, assuming a zero baseline. In this case the attribution for $x_1$ is larger than that for $x_2$, an opposite rank with respect to the results of the local model. Notice that we used nothing but Gradient * Input as global attribution method which, in the linear case, is equivalent to all other methods analyzed above.

The methods listed in Table 1 are examples of global attribution methods. Although local attribution methods are not further discussed here, we can mention Saliency maps (Simonyan et al., 2014) as an example. In fact, Montavon et al. (2017) showed that Saliency maps can be seen as the first-order term of a Taylor decomposition of the function implemented by the network, computed at a point *infinitesimally close* to the actual input.

Finally, we notice that global and local attributions accomplish two different tasks, that only converge when the model is linear. Local attributions aim to explain how the input should be changed in order to obtain a desired variation on the output. One practical application is the generation of adversarial perturbations, where genuine input samples are minimally perturbed to cause a disruptive change in the output (Szegedy et al., 2014; Goodfellow et al., 2015). On the contrary, global attributions should be used to identify the marginal effect that the presence of a feature has on the output, which is usually desirable from an explanation method.

## 4    EVALUATING ATTRIBUTIONS

Attributions methods are hard to evaluate empirically because it is difficult to distinguish errors of the model from errors of the attribution method explaining the model (Sundararajan et al., 2017). For this reason the final evaluation is often qualitative, based on the inspection of the produced attribution maps. We argue, however, that this introduces a strong bias in the evaluation: as humans, one would judge more favorably methods that produce explanations closer to his own expectations, at the cost of penalizing those methods that might more closely reflect the network behavior. In order to develop better quantitative tools for the evaluation of attribution methods, we first need to define the goal that an ideal attribution method should achieve, as different methods might be suitable for different tasks (Subsection 3.2).

Consider the attribution maps on MNIST produced by a CNN that uses Sigmoid nonlinearities (Figure 3a-b). Integrated Gradients assigns high attributions to the background space in the middle of the image, while Occlusion-1 does not. One might be tempted to declare Integrated Gradients a better attribution method, given that the heatmap is less scattered and that the absence of strokes in the middle of the image might be considered a good clue in favor of a zero digit. In order to evaluate the hypothesis, we apply a variation of the *region perturbation* method (Samek et al., 2016) removing pixels according to the ranking provided by the attribution maps (higher first (+) or lower first (-)). We perform this operation replacing one pixel at the time with a zero value and measuring the variation in the target activation. The results in Figure 3c show that pixels highlighted by Occlusion-1 initially have a higher impact on the target output, causing a faster variation from the initial value. After removing about 20 pixels or more, Integrated Gradients seems to detect more relevant features, given that the variation in the target output is stronger than for Occlusion-1.

This is an example of attribution methods solving two different goals: we argue that while Occlusion-1 is better explaining the role of each feature considered in isolation, Integrated Gradients is better in capturing the effect of multiple features together. It is possible, in fact, that given the presence of several white pixels in the central area, the role of each one alone is not prominent, while the deletion of several of them together causes a drop in the output score. In order to test this assumption systematically, we propose a property called *Sensitivity-n*.

**Sensitivity-n**. *An attribution method satisfies Sensitivity-n when the sum of the attributions for any subset of features of cardinality $n$ is equal to the variation of the output $S_c$ caused removing the*

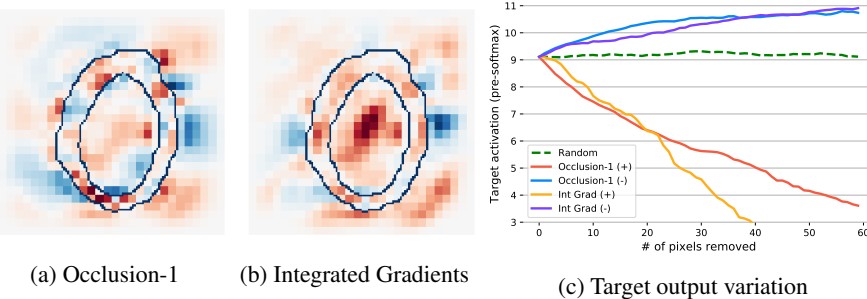

(a) Occlusion-1  (b) Integrated Gradients  (c) Target output variation

Figure 3: Comparison of attribution maps and (a-b) and plot of target output variation as some features are removed from the input image. Best seen in electronic form.

*features in the subset. Mathematically when, for all subsets of features $x_S = [x_1, ... x_n] \subseteq x$, it holds $\sum_{i=1}^{n} R_i^c(x) = S_c(x) - S_c(x_{[x_S=0]})$.*

When $n = N$, with $N$ being the total number of input features, we have $\sum_{i=0}^{N} R_i^c(x) = S_c(x) - S_c(\bar{x})$, where $\bar{x}$ is an input baseline representing an input from which all features have been removed. This is nothing but the definition of *Completeness* or *Summation to Delta*, for which Sensitivity-$n$ is a generalization. Notice that Occlusion-1 satisfy Sensitivity-1 by construction, like Integrated Gradients and DeepLIFT satisfy Sensitivity-N (the latter only without multiplicative units for the reasons discussed in Section 3.1). $\epsilon$-LRP satisfies Sensitivity-N if the conditions of Proposition 3-(ii) are met. However no methods in Table 1 can satisfy Sensitivity-n for all $n$:

**Proposition 4.** *All attribution methods defined in Table 1 satisfy Sensitivity-n for all values of $n$ if and only if applied to a linear model or a model that behaves linearly for a selected task. In this case, all methods of Table 1 are equivalent.*

The proof of Proposition 4 is provided in Appendix A.3. Intuitively, if we can only assign a scalar attribution to each feature, there are not enough degrees of freedom to capture nonlinear interactions. Besides degenerate cases when DNNs behave as linear systems on a particular dataset, the attribution methods we consider can only provide a partial explanation, sometimes focusing on different aspects, as discussed above for Occlusion-1 and Integrated Gradients.

## 4.1 MEASURING SENSITIVITY

Although no attribution method satisfies Sensitivity-$n$ for all values of $n$, we can measure how well the sum of the attributions $\sum_{i=1}^{N} R_i^c(x)$ and the variation in the target output $S_c(x) - S_c(x_{[x_S=0]})$ correlate on a specific task for different methods and values of $n$. This can be used to compare the behavior of different attribution methods.

While it is intractable to test all possible subsets of features of cardinality $n$, we estimate the correlation by randomly sampling one hundred subsets of features from a given input $x$ for different values of $n$. Figure 4 reports the Pearson correlation coefficient (PCC) computed between the sum of the attributions and the variation in the target output varying $n$ from one to about 80% of the total number of features. The PCC is averaged across a thousand of samples from each dataset. The sampling is performed using a uniform probability distribution over the features, given that we assume no prior knowledge on the correlation between them. This allows to apply this evaluation not only to images but to any kind of input.

We test all methods in Table 1 on several tasks and different architectures. We use the well-known MNIST dataset (LeCun et al., 1998) to test how the methods behave with two different architectures (a Multilayer Perceptron (MLP) and a CNN) and four different activation functions. We also test a simple CNN for image classification on CIFAR10 (Krizhevsky & Hinton, 2009) and the more complex Inception V3 architecture (Szegedy et al., 2016) on ImageNet (Russakovsky et al., 2015) samples. Finally, we test a model for sentiment classification from text data. For this we use the IMDB dataset (Maas et al., 2011), applying both a MLP and an LSTM model. Details about the architectures can be found in Appendix C. Notice that it was not our goal, nor a requirement, to reach

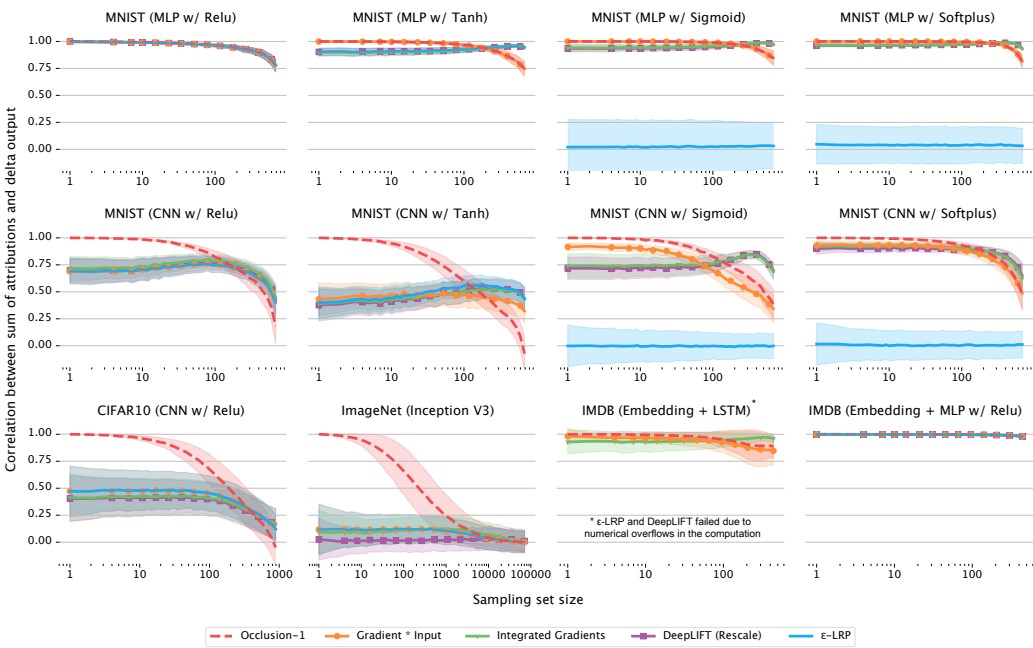

Figure 4: Test of Sensitivity-$n$ for several values of $n$, over different tasks and architectures.

the state-of-the-art in these tasks since attribution methods should be applicable to any model. On the contrary, the simple model architecture used for sentiment analysis enables us to show a case where a DNN degenerates into a nearly-linear behavior, showing in practice the effects of Proposition 4. From these results we can formulate some considerations:

1. **Input might contain negative evidence.** Since all methods considered produce signed attributions and the correlation is close to one for at least some value of $n$, we conclude that the input samples can contain negative evidence and that it can be correctly reported. This conclusion is further supported by the results in Figure 3c where the occlusion of negative evidence produces an *increase* in the target output. On the other hand, on complex models like Inception V3, all gradient-based methods show low accuracy in predicting the attribution sign, leading to heatmaps affected by high-frequency noise (Figure 2).

2. **Occlusion-1 better identifies the few most important features.** This is supported by the fact that Occlusion-1 satisfies Sensitivity-1, as expected, while the correlation decreases monotonically as $n$ increases in all our experiments. For simple models, the correlation remains rather high even for medium-size sets of pixels but Integrated Gradients, DeepLIFT and LRP should be preferred when interested in capturing global nonlinear effects and cross-interactions between different features. Notice also that Occlusion-1 is much slower than gradient-based methods.

3. In some cases, like in MNIST-MLP w/ Tanh, **Gradient * Input approximates the behavior of Occlusion-1 better than other gradient-based methods.** This suggests that the instant gradient computed by Gradient * Input is feature-wise very close to the average gradient for these models.

4. **Integrated Gradients and DeepLIFT have very high correlation**, suggesting that the latter is a good (and faster) approximation of the former in practice. This does not hold in presence of multiplicative interactions between features (eg. IMDB-LSTM). In these cases the analyzed formulation of DeepLIFT should be avoided for the reasons discussed in Section 3.1.

5. $\epsilon$-**LRP is equivalent to Gradient * Input when all nonlinearities are ReLUs, while it fails when these are Sigmoid or Softplus.** When the nonlinearities are such that $f(0) \neq 0$, $\epsilon$-LRP diverges from other methods, cannot be seen as a discrete gradient approximator and may lead to numerical instabilities for small values of the stabilizer (Section 3.1). It has been shown, however, that adjusting the propagation rule for multiplicative interactions and avoiding critical nonlinearities, $\epsilon$-LRP can be applied to LSTM networks, obtaining interesting results (Arras et al., 2017).

Unfortunately, these changes obstacle the formulation as modified chain-rule and make *ad-hoc* implementation necessary.

6. **All methods are equivalent when the model behaves linearly.** On IMDB (MLP), where we used a very shallow network, all methods are equivalent and the correlation is maximum for almost all values of $n$. From Proposition 4 we can say that the model approximates a linear behavior (each word contributes to the output independently from the context).

## 5 CONCLUSIONS

In this work, we have analyzed Gradient * Input, $\epsilon$-LRP, Integrated Gradients and DeepLIFT (Rescale) from theoretical and practical perspectives. We have shown that these four methods, despite their apparently different formulation, are strongly related, proving conditions of equivalence or approximation between them. Secondly, by reformulating $\epsilon$-LRP and DeepLIFT (Rescale), we have shown how these can be implemented as easy as other gradient-based methods. Finally, we have proposed a metric called *Sensitivity-n* which helps to uncover properties of existing attribution methods but also traces research directions for more general ones.

ACKNOWLEDGEMENTS

This work was partially funded by the Swiss Commission for Technology and Innovation (CTI Grant No. 19005.1 PFES-ES). We would like to thank Brian McWilliams and David Tedaldi for their helpful feedback.

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

# A    PROOF OF PROPOSITIONS

## A.1    PROOF OF PROPOSITION 1

For the following proof we refer to the $\epsilon$ propagation rule defined as in Equation 56 of Bach et al. (2015). According to this definition the bias terms can be assigned part of the relevance. We also assume the stabilizer term $\epsilon \cdot sign(\sum_{i'}(z_{i'j} + b_j))$ at the denominator of Equation 2 is small enough to be neglected, which is anyway necessary for the property of relevance conservation to hold.

*Proof.* We proceed by induction. By definition, the $\epsilon$-LRP relevance of the target neuron $c$ on the top layer $L$ is defined to be equal to the output of the neuron itself, $S_c$:

$$r_c^{(L)} = S_c(x) = f\left(\sum_j w_{cj}^{(L,L-1)} x_j^{(L-1)} + b_c\right) \tag{6}$$

The relevance of the parent layer is:

$$r_j^{(L-1)} = r_c^L \frac{w_{cj}^{(L,L-1)} x_j^{(L-1)}}{\sum_{j'} w_{cj'}^{(L,L-1)} x_{j'}^{(L-1)} + b_c} \qquad \blacktriangleright \text{LRP prop. rule (Eq. 2)}$$

$$= f\left(\sum_{j'} w_{cj'}^{(L,L-1)} x_{j'}^{(L-1)} + b_c\right) \frac{w_{cj}^{(L,L-1)} x_j^{(L-1)}}{\sum_{j'} w_{cj'}^{(L,L-1)} x_{j'}^{(L-1)} + b_c} \qquad \blacktriangleright \text{replacing Eq. 6}$$

$$= g^{LRP}\left(\sum_{j'} w_{cj'}^{(L,L-1)} x_{j'}^{(L-1)} + b_c\right) w_{cj}^{(L,L-1)} x_j^{(L-1)} \qquad \blacktriangleright \text{by definition of } g^{LRP}$$

$$= \frac{\partial^{g^{LRP}} S_c(x)}{\partial x_j^{(L-1)}} x_j^{(L-1)} \qquad \blacktriangleright \text{by definition of } \partial^g \text{ (Eq. 5)}$$

For the inductive step we start from the hypothesis that on a generic layer $l$ the LRP explanation is:

$$r_i^{(l)} = \frac{\partial^{g^{LRP}} S_c(x)}{\partial x_i^{(l)}} x_i^{(l)} \tag{7}$$

then for layer $l - 1$ it holds:

$$r_j^{(l-1)} = \sum_i r_i^{(l)} \frac{w_{ij}^{(l,l-1)} x_j^{(l-1)}}{\sum_{j'} w_{ij'}^{(l,l-1)} x_{j'}^{(l-1)} + b_i} \qquad \blacktriangleright \text{LRP propagation rule (Eq. 2)}$$

$$= \sum_i \frac{\partial^{g^{LRP}} S_c(x)}{\partial x_i^{(l)}} \underbrace{\frac{x_i^{(l)}}{\sum_{j'} w_{ij'}^{(l,l-1)} x_{j'}^{(l-1)} + b_i}}_{g^{LRP}} w_{ij}^{(l,l-1)} x_j^{(l-1)} \qquad \blacktriangleright \text{replacing Eq. 7}$$

$$= \frac{\partial^{g^{LRP}} S_c(x)}{\partial x_j^{(l-1)}} x_j^{(l-1)} \qquad \blacktriangleright \text{chain-rule for } \partial^g$$

In the last step, we used the chain-rule for $\partial^g$, defined in Equation 5. This only differs from the gradient chain-rule by the fact that the gradient of the nonlinearity $f'$ between layers $l - 1$ and $l$ is replaced with the value of $g^{LRP}$, ie. the ratio between the output and the input at the nonlinearity.

$\square$

## A.2   Proof of Proposition 2

Similarly to how a chain rule for gradients is constructed, DeepLIFT computes a multiplicative term, called "multiplier", for each operation in the network. These terms are chained to compute a global multiplier between two given units by summing up all possible paths connecting them. The chaining rule, called by the authors "chain rule for multipliers" (Eq. 3 in (Shrikumar et al., 2017)) is identical to the chain rule for gradients, therefore we only need to prove that the multipliers are equivalent to the terms used in the computation of our modified backpropagation.

*Linear operations.* For Linear and Convolutional layers implementing operations of the form $z_j = \sum_i (w_{ji} \cdot x_i) + b_j$, the DeepLIFT multiplier is defined to be $m = w_{ji}$ (Sec. 3.5.1 in (Shrikumar et al., 2017)). In our formulation the gradient of linear operations is not modified, hence it is $\partial z_i / \partial x_i = w_{ji}$, equal to the original DeepLIFT multiplier.

*Nonlinear operations.* For a nonlinear operation with a single input of the form $x_i = f(z_i)$ (i.e. any nonlinear activation function), the DeepLIFT multiplier (Sec. 3.5.2 in Shrikumar et al. (Shrikumar et al., 2017)) is:

$$m = \frac{\Delta x}{\Delta z} = \frac{f(z_i) - f(\bar{z}_i)}{z_i - \bar{z}_i} = g^{DL} \tag{8}$$

Nonlinear operations with multiple inputs (eg. 2D pooling) are not addressed in (Shrikumar et al., 2017). For these, we keep the original operations' gradient unmodified as in the DeepLIFT public implementation. [3]

### A.3 PROOF OF PROPOSITION 4

By linear model we refer to a model whose target output can be written as $S_c(x) = \sum_i h_i(x_i)$, where all $h_i$ are compositions of linear functions. As such, we can write

$$S_c(x) = \sum_i a_i x_i + b_i \tag{9}$$

for some some $a_i$ and $b_i$. If the model is linear only in the restricted domain of a task inputs, the following considerations hold in the domain. We start the proof by showing that, on a linear model, all methods of Table 1 are equivalent.

*Proof.* In the case of Gradient * Input, on a linear model it holds $R_i^c(x) = x_i \cdot \frac{\partial S_c(x)}{\partial x_i} = x_i h_i'(x) = a_i x_i$, being all other derivatives in the summation zero. Since we are considering a linear model, all nonlinearities $f$ are replaced with the identity function and therefore $\forall z : g^{DL}(z) = g^{LRP}(z) = f'(z) = 1$ and the modified chain-rules for LRP and DeepLIFT reduce to the gradient chain-rule. This proves that $\epsilon$-LRP and DeepLIFT with a zero baseline are equivalent to Gradient * Input in the linear case. For Integrated Gradients the gradient term is constant and can be taken out of the integral: $R_i^c(x) = x_i \cdot \int_{\alpha=0}^1 \frac{\partial S_c(\tilde{x})}{\partial(\tilde{x}_i)}\big|_{\tilde{x}=\bar{x}+\alpha(x-\bar{x})} d\alpha = x_i \cdot \int_{\alpha=0}^1 h_i'(\alpha x_i) d\alpha = a_i \cdot \int_{\alpha=0}^1 d\alpha = a_i x_i$. Finally, for Occlusion-1, by the definition we get $R_i^c(x) = S_c(x) - S_c(x_{[x_i=0]}) = \sum_j (a_j x_j + b_j) - \sum_{j\neq i}(a_j x_j + b_j) - b_i = a_i x_i$, which completes the proof the proof of equivalence for the methods in Table 1 in the linear case.

If we now consider any subset of $n$ features $x_S \subseteq x$, we have for Occlusion-1:

$$\sum_{i=1}^n R_i^c(x) = \sum_{i=1}^n (a_i x_i) = S_c(x) - S_c(x_{[x_S=0]}) \tag{10}$$

where the last equality holds because of the definition of linear model (Equation 9). This shows that Occlusion-1, and therefore all other equivalent methods, satisfy Sensitivity-$n$ for all $n$ if the model is linear. If, on the contrary, the model is not linear, there must exists two features $x_i$ and $x_j$ such that $S_c(x) - S_c(x_{[x_i=0;x_j=0]}) \neq 2 \cdot S_c(x) - S_c(x_{[x_i=0]}) - S_c(x_{[x_j=0]})$. In this case, either Sensitivity-1 or Sensitivity-2 must be violated since all methods assign a single attribution value to $x_i$ and $x_j$. $\square$

## B ABOUT THE NEED FOR A BASELINE

In general, a non-zero attribution for a feature implies the feature is expected to play a role in the output of the model. As pointed out by Sundararajan et al. (2017), humans also assign blame to a cause by comparing the outcomes of a process including or not such cause. However, this requires

---

[3]DeepLIFT public repository: `https://github.com/kundajelab/deeplift`. Retrieved on 25 Sept. 2017

the ability to test a process with and without a specific feature, which is problematic with current neural network architectures that do not allow to explicitly remove a feature without retraining. The usual approach to *simulate* the absence of a feature consists of defining a baseline $x'$, for example the black image or the zero input, that will represent absence of information. Notice, however, that the baseline must necessarily be chosen in the domain of the input space and this creates inherently an ambiguity between a valid input that incidentally assumes the baseline value and the placeholder for a missing feature. On some domains, it is also possible to marginalize over the features to be removed in order to simulate their absence. Zintgraf et al. (2017) showed how local coherence of images can be exploited to marginalize over image patches. Unfortunately, this approach is extremely slow and only provide marginal improvements over a pre-defined baseline. What is more, it can only be applied to images, where contiguous features have a strong correlation, hence our decision to use the method by Zeiler & Fergus (2014) as our benchmark instead.

When a baseline value has to be defined, zero is the canonical choice (Sundararajan et al., 2017; Zeiler & Fergus, 2014; Shrikumar et al., 2017). Notice that Gradient * Input and LRP can also be interpreted as using a zero baseline implicitly. One possible justification relies on the observation that in network that implements a chain of operations of the form $z_j = f(\sum_i (w_{ji} \cdot z_i) + b_j)$, the all-zero input is somehow neutral to the output (ie. $\forall c \in C : S_c(0) \approx 0$). In fact, if all additive biases $b_j$ in the network are zero and we only allow nonlinearities that cross the origin, the output for a zero input is exactly zero for all classes. Empirically, the output is often near zero even when biases have different values, which makes the choice of zero for the baseline reasonable, although arbitrary.

## C  EXPERIMENTS SETUP

### C.1  MNIST

The MNIST dataset (LeCun et al., 1998) was pre-processed to normalize the input images between -1 (background) and 1 (digit stroke). We trained both a DNN and a CNN, using four activation functions in order to test how attribution methods generalize to different architectures. The lists of layers for the two architectures are listed below. The activations functions are defined as $ReLU(x) = max(0, x)$, $Tanh(x) = sinh(x)/cosh(x)$, $Sigmoid(x) = 1/(1 + e^{-x})$ and $Softplus(x) = ln(1 + e^x)$ and have been applied to the output of the layers marked with $\dagger$ in the tables below. The networks were trained using Adadelta (Zeiler, 2012) and early stopping. We also report the final test accuracy.

| MNIST MLP |
| --- |
| Dense$^\dagger$ (512) |
| Dense$^\dagger$ (512) |
| Dense (10) |

| MNIST CNN |
| --- |
| Conv 2D$^\dagger$ (3x3, 32 kernels) |
| Conv 2D$^\dagger$ (3x3, 64 kernels) |
| Max-pooling (2x2) |
| Dense$^\dagger$ (128) |
| Dense (10) |

| Test set accuracy (%) | | |
| --- | --- | --- |
| | MLP | CNN |
| ReLU | 97.9 | 99.1 |
| Tanh | 98.1 | 98.8 |
| Sigmoid | 98.1 | 98.6 |
| Softplus | 98.1 | 98.8 |

### C.2  CIFAR-10

The CIFAR-10 dataset (Krizhevsky & Hinton, 2009) was pre-processed to normalized the input images in range [-1; 1]. As for MNIST, we trained a CNN architecture using Adadelta and early stopping. For this dataset we only used the $ReLU$ nonlinearity, reaching a final test accuracy of 80.5%. For gradient-based methods, the attribution of each pixel was computed summing up the attribution of the 3 color channels. Similarly, Occlusion-1 was performed setting all color channels at zero at the same time for each pixel being tested.

| CIFAR-10 CNN |
|:---:|
| Conv 2D$^\dagger$ (3x3, 32 kernels) |
| Conv 2D$^\dagger$ (3x3, 32 kernels) |
| Max-pooling (2x2) |
| Dropout (0.25) |
| Conv 2D$^\dagger$ (3x3, 64 kernels) |
| Conv 2D$^\dagger$ (3x3, 64 kernels) |
| Max-pooling (2x2) |
| Dropout (0.25) |
| Dense$^\dagger$ (256) |
| Dropout (0.5) |
| Dense (10) |

## C.3 INCEPTION V3

We used a pre-trained Inception V3 network. The details of this architecture can be found in Szegedy et al. (2016). We used a test dataset of 1000 ImageNet-compatible images, normalized in [-1; 1] that was classified with 95.9% accuracy. When computing attributions, the color channels were handled as for CIFAR-10.

## C.4 IMDB

We trained both a shallow MLP and an LSTM network on the IMDB dataset (Maas et al., 2011) for sentiment analysis. For both architectures, we trained a small embedding layer considering only the 5000 most frequent words in the dataset. We also limited the maximum length of each review to 500 words, padding shorter ones when necessary. We used $ReLU$ nonlinearities for the hidden layers and trained using Adam (Kingma & Ba, 2014) and early stopping. The final test accuracy is 87.3% on both architectures. For gradient-based methods, the attribution of each word was computed summing up the attributions over the embedding vector components corresponding to the word. Similarly, Occlusion-1 was performed setting all components of the embedding vector at zero for each word to be tested.

| IMDB MLP | IMDB LSTM |
|:---:|:---:|
| Embedding (5000x32) | Embedding (5000x32) |
| Dense (250) | LSTM (64) |
| Dense (1) | Dense (1) |

