# OpenReview forum: "Towards better understanding of gradient-based attribution methods for Deep Neural Networks"
_ICLR.cc/2018/Conference — Accept (Poster)_

### Official Review · AnonReviewer2 · 2017-11-27
**Ties together attribution methods in a unifying and systematic way**

**Rating:** 7
**Confidence:** 3

**Review:**

This paper discusses several gradient based attribution methods, which have been popular for the fast computation of saliency maps for interpreting deep neural networks. The paper provides several advances:
- \epsilon-LRP and DeepLIFT are formulated in a way that can be calculated using the same back-propagation as training.
- This gives a more unified way of understanding, and implementing the methods.
- The paper points out situations when the methods are equivalent
- The paper analyses the methods' sensitivity to identifying single and joint regions of sensitivity
- The paper proposes a new objective function to measure joint sensitivity

Overall, I believe this paper to be a useful contribution to the literature. It both solidifies understanding of existing methods and provides new insight into quantitate ways of analysing methods. Especially the latter will be appreciated.

---

> ### Author Response · Authors · 2017-12-19
> **Re: Ties together attribution methods in a unifying and systematic way**
>
> We thank the reviewer for his/her feedbacks.

---

### Official Review · AnonReviewer3 · 2017-11-27
**Good paper, but discussion on methods which do not fit into the proposed framework could be extended.**

**Rating:** 6
**Confidence:** 5

**Review:**

The paper summarizes and compares some of the current explanation techniques for deep neural networks that rely on the redistribution of relevance / contribution values from the output to the input space.

The main contributions are the introduction of a unified framework that expresses 4 common attribution techniques (Gradient * Input, Integrated Gradient, eps-LRP and DeepLIFT) in a similar way as modified gradient functions and the definition of a new evaluation measure ('sensitivity n') that generalizes the earlier defined properties of 'completeness' and 'summation to delta'.

The unified framework is very helpful since it points out equivalences between the methods and makes the implementation of eps-LRP and DeepLIFT substantially more easy on modern frameworks. However, as correctly stated by the authors some of the unification (e.g. relation between LRP and Gradient*Input) has been already mentioned in prior work.

Sensitivity-n as a measure tries to tackle the difficulty of estimating the importance of features that can be seen either separately or in combination. While the measure shows interesting trends towards a linear behaviour for simpler methods, it does not persuade me as a measure of how well the relevance attribution method mimics the decision making process and does not really point out substantial differences between the different methods. Furthermore, The authors could comment on the relation between sensitivity-n and region perturbation techniques (Samek et al., IEEE TNNLS, 2017). Sensitivtiy-n seems to be an extension of the region perturbation idea to me.

It would be interesting to see the relation between the "unified" gradient-based explanation methods and approaches (e.g. Saliency maps, alpha-beta LRP, Deep Taylor, Deconvolution Networks, Grad-CAM, Guided Backprop ...) which do not fit into the unification framework. It's good that the author mention these works, still it would be great to see more discussion on the advantages/disadvantages, because these methods may have some nice theoretically properties (see e.g. the discussion on gradient vs. decompositiion techniques in Montavon et al., Digital Signal Processing, 2017) which can not be incorporated into the unified framework.

---

> ### Author Response · Authors · 2017-12-19
> **Re: Good paper**
>
> Thanks for your extensive review and useful feedbacks.
>
> While we agree that it would be interesting to compare with other mentioned methods, the reasons we decided not to do so are various. Saliency maps and Deep Taylor Decomposition only produce positive attribution maps and and this would penalize these methods in the sensitivity-n metric given that our task inputs do contain some negative evidence (as shown in Figure 3c). Similarly, alpha-beta LRP also adds some bias towards positive attributions with the parameters suggested by the authors. Grad-CAM, Deconvolutional Networks and Guided Backpropagation can only be applied to specific network architectures and do not fit our goal to compare methods across tasks and architectures, while we believe it is important for attribution methods to be as general as possible. We reported the same arguments at the end of section 2.2.
> We also agree with your statement that other methods have been shown to have interesting theoretical properties. We actually did not intend to claim the superiority of gradient-based methods and added a note to clarify this in Section 3.1 in our last revision.
>
> About the connection with the region perturbation technique (Samek et al. 2017), this is similar to what we use (and now mention explicitly) to produce Figure 3c, with the difference that we occlude one pixel at the time, we produce the curves for the negative ranking as well as for the positive and we plot directly the output variation on the y-axis instead of the AOPC. This technique evaluates methods based on i) how "fast" the target activation drops or increase and ii) how "much" the target activation changes. However, we show in Figure 3c that these two criteria often collide if the curves for different methods intersect. This is in fact what motivated sensitivity-n.
> With sensitivity-n, we fix a value n and remove random subsets of n features from the input (without following the ranking given by the attribution maps) and measure the Pearson correlation with the output variation. This shows that different methods are better at producing different explanations (influence of single features vs influence of regions) and therefore the question itself of which is the best attribution method does not make sense if a task is not further specified.

---

### Official Review · AnonReviewer1 · 2017-11-29
**Useful work showing the similarity between different neural network interpretation techniques**

**Rating:** 7
**Confidence:** 4

**Review:**

The paper shows that several recently proposed interpretation techniques for neural network are performing similar processing and yield similar results. The authors show that these techniques can all be seen as a product of input activations and a modified gradient, where the local derivative of the activation function at each neuron is replaced by some fixed function.

A second part of the paper looks at whether explanations are global or local. The authors propose a metric called sensitivity-n for that purpose, and make some observations about the optimality of some interpretation techniques with respect to this metric in the linear case. The behavior of each explanation w.r.t. these properties is then tested on multiple DNN models tested on real-world datasets. Results further outline the resemblance between the compared methods.

In the appendix, the last step of the proof below Eq. 7 is unclear. As far as I can see, the variable g_i^LRP wasn’t defined, and the use of Eq. 5 to achieve this last could be better explained. There also seems to be some issues with the ordering i,j, where these indices alternatively describe the lower/higher layers, or the higher/lower layers.

---

> ### Author Response · Authors · 2017-12-19
> **Proof improved for clarity in revision 2**
>
> Thanks for your your review.
> We reworked the last part of the proof A.1 in our last revision. In particular, we removed the variable g_i that was not defined and better explained the last step.
> We did not find issues with the ordering of the subscripts i,j in the proof itself but we did notice that it was inconsistent with the convention we used in Section 2. We have now fixed it such that, when two subscripts are present, the first one always refers to the layer closer to the output.

---

### Public Comment · ~Ruoyu_Chen2 · 2023-12-15
**Question about your paper**

Dear Authors:

I'm reading your ICLR18 paper: "Towards better understanding of gradient-based attribution methods for deep neural networks" and have one question. In your paper, you mentioned, "we estimate the correlation by randomly sampling one hundred subsets of features from a given input x for different values of $n$.". Does this mean sampling 100 subsets with different $n$ for each subset, or collecting 100 subsets to calculate PPC when $n$ is constant?
Looking forward to your reply! Thanks!

---

### Decision · Program_Chairs · 2018-01-29
**ICLR 2018 Conference Acceptance Decision**

**Decision:**

Accept (Poster)

**Comment:**

With scores of 7-7-6  and the justification below the AC recommends acceptance.

One of the reviewers summarizes why this is a good paper as follows:

"This paper discusses several gradient based attribution methods, which have been popular for the fast computation of saliency maps for interpreting deep neural networks. The paper provides several advances:
- This gives a more unified way of understanding, and implementing the methods.
- The paper points out situations when the methods are equivalent
- The paper analyses the methods' sensitivity to identifying single and joint regions of sensitivity
- The paper proposes a new objective function to measure joint sensitivity"